# Adding Value in Production of Multifunctional Polylactide (PLA)–ZnO Nanocomposite Films through Alternative Manufacturing Methods

**DOI:** 10.3390/molecules26072043

**Published:** 2021-04-02

**Authors:** Marius Murariu, Samira Benali, Yoann Paint, Anne-Laure Dechief, Oltea Murariu, Jean-Marie Raquez, Philippe Dubois

**Affiliations:** 1Laboratory of Polymeric and Composite Materials, Materia Nova Materials R&D Center & UMons Innovation Center, Avenue Copernic 3, 7000 Mons, Belgium; yoann.paint@materianova.be (Y.P.); anne-laure.dechief@materianova.be (A.-L.D.); oltea.murariu@materianova.be (O.M.); 2Center of Innovation and Research in Materials and Polymers (CIRMAP), Laboratory of Polymeric and Composite Materials, University of Mons (UMons), Place du Parc 20, 7000 Mons, Belgium; samira.benali@umons.ac.be (S.B.); jeanmarie.raquez@umons.ac.be (J.-M.R.)

**Keywords:** biopolymers, poly(lactic acid), PLA, nanocomposites, zinc oxide, masterbatch (MB), melt-compounding, film extrusion, packaging, thermal and mechanical characterizations, UV-vis spectroscopy, multifunctional properties

## Abstract

Due to the added value conferred by zinc oxide (ZnO) nanofiller, e.g., UV protection, antibacterial action, gas-barrier properties, poly(lactic acid) (PLA)–ZnO nanocomposites show increased interest for utilization as films, textile fibers, and injection molding items. The study highlights the beneficial effects of premixing ZnO in PLA under given conditions and its use as masterbatch (MB), a very promising alternative manufacturing technique. This approach allows reducing the residence time at high processing temperature of the thermo-sensitive PLA matrix in contact of ZnO nanoparticles known for their aptitude to promote degradation effects onto the polyester chains. Various PLA–ZnO MBs containing high contents of silane-treated ZnO nanoparticles (up to 40 wt.% nanofiller specifically treated with triethoxycaprylylsilane) were produced by melt-compounding using twin-screw extruders. Subsequently, the selected MBs were melt blended with pristine PLA to produce nanocomposite films containing 1–3 wt.% ZnO. By comparison to the more traditional multi-step process, the MB approach allowed the production of nanocomposites (films) having improved processing and enhanced properties: PLA chains displaying higher molecular weights, improved thermal stability, fine nanofiller distribution, and thermo-mechanical characteristic features, while the UV protection was confirmed by UV-vis spectroscopy measurements. The MB alternative is viewed as a promising flexible technique able to open new perspectives to produce more competitive multifunctional PLA–ZnO nanocomposites.

## 1. Introduction

After more than 20 years of research, poly(lactic acid) or polylactide (PLA), an aliphatic polyester produced from renewable resources, represents the key candidate biopolymer for further developments and the rise of global manufacturing capacities [1,2,3,4,5]. PLA production begins with the fermentation of polysaccharides or sugar, e.g., extracted from corn or sugar beet, and corresponding wastes, while the most relevant end-life scenario is related to its biodegradability under controlled industrial composting conditions. Due to very interesting properties (particularly, great transparency, high tensile strength, and rigidity/Young’s modulus, good flexural strength), PLA is currently receiving considerable attention for traditional applications such as packaging, as well as production of textile fibers, and it also finds higher added value for technical and biomedical applications [6,7,8,9,10]. The last tendencies show that improved PLA products and new grades are required for different markets and applications (packaging, transportation, electronic and electrical devices, mechanical, and automotive parts, etc.). Consequently, for a larger utilization, the profile of PLA properties is tuned up by combining the polyester matrix with reinforcing fibers, micro- and nano-fillers, flame retardants, impact modifiers, plasticizers, and other additives of specialty [3]. Nowadays, the addition into PLA matrix of selected nanofillers, e.g., organo-modified layered silicates (OMLS), silver, zinc oxide, graphite derivatives, carbon nanotubes (CNT), carbon black, etc., is considered as a modern approach that can lead to major improvements of PLA characteristics (mechanical, thermal, barrier, etc.). Furthermore, these nanocomposites are characterized by specific end-use properties such as anti-UV and anti-bacterial protection, antistatic to conductive electrical characteristics, enhanced wear resistance, higher speed crystallization, fire-retardancy, and so on [4,11,12,13,14,15,16,17,18,19].

Zinc oxide (ZnO) is a well-known environmentally friendly and multifunctional inorganic nanofiller, with an optical band gap in the UV region that makes it useful as an efficient UV-light absorber. The ZnO nanofillers can be mixed with different polymers to produce nanocomposites characterized by a large array of properties such as effective antibacterial protection, intensive ultraviolet absorption, or other characteristic features [20,21,22,23]. Interestingly, ZnO nanoparticles (NPs) and Zn-based derivatives are known as very efficient catalysts in the synthesis of PLA through ring-opening polymerization (ROP) of lactide, as well as in “unzipping” PLA depolymerization, allowing its recycling and the recovering of initial monomers [24,25]. Indeed, within the first studies it was stated out by us that the addition of untreated ZnO NPs into PLA, at melt-processing temperature, led to the severe degradation of the polyester matrix, a process well-determined by ZnO loading. Likewise, the same conclusion was also shared by other research groups [26,27]. However, by considering the high effectiveness of Zn-based products (such as untreated ZnO) to trigger PLA degradation [28], other researchers have preferred to use the solvent casting method to produce PLA–ZnO nanocomposites [16,29].

On the other hand, following the interest for the use of these products on a larger scale, it was first reported by our group the possibility to produce PLA–ZnO nanocomposites via melt-blending technology and using ZnO NPs previously surface-treated with a specific silane (i.e., triethoxycaprylylsilane) to limit polyester degradation [26,30,31]. These nanocomposites produced using a method easy transferable at larger scale, have been tested in the production of fibers, films, or other materials, showing multifunctional end-use properties (UV protection, antibacterial, barrier, etc.) [26,30,32,33]. In a subsequent study, epoxy-functional styrene-acrylic oligomers have been proposed as effective chain extenders (CE) to improve the properties of PLA–ZnO nanocomposites (molecular, thermal, rheological, etc.) while maintaining their specific end-use characteristics [34]. Still, by comparing the different approaches described as ‘state-of-the-art’, we assume that the melt-blending using twin-screw extruders has larger industrial applicability with respect to other techniques, such as the solvent-casting method.

Regarding the utilization of ZnO as nanofiller for PLA, it is noteworthy mentioning that the surface treatment of NPs is a prerequisite to limit the degradation of the polyester matrix, which is determined by the loading of ZnO [26,27]. In the real case of industrial applications, and especially at high processing temperature and long residence time, an increased stability is required by end-users to produce films, fibers, injection molded parts, etc. since the decrease of molecular parameters can affect many key properties (e.g., melt-fluidity, thermal and mechanical characteristics). Therefore, the control of thermal degradation and rheology of these nanocomposites represents a major challenge, whereas novel manufacturing techniques are of practical concern. We will mention that in the traditional multi-step process, involving the melt-mixing of PLA with ZnO nanofillers using twin-screw extruders to produce nanocomposites (as granules) in the first phase, followed by a second processing to obtain fibers, films, etc., it is very difficult to prevent the degradation of polyester matrix. Moreover, because these nanocomposites are of great practical consideration [26,35,36,37], there is a need to find the most adapted techniques to produce more performant products. Spectacular improvements can be obtained by limiting the residence time at high temperature of PLA in the presence of ZnO. Unfortunately, according to the best of our knowledge, the potential of the masterbatch (MB) technique was not enough considered to produce these specific nanocomposites. Furthermore, the utilization of MBs could represent a more flexible way to obtain nanocomposites having different loadings of ZnO, improved processing and thermo-mechanical properties, better preservation of the molecular weights. The main objective of this study has been to highlight the beneficial effects of MB addition on the performances of PLA–ZnO nanocomposite films as final products, while other applications can be also concerned. Moreover, this is one of the first works dealing with the production and properties of PLA–ZnO MBs, while the characteristic features of final nanocomposites extruded as films using highly filled MBs are directly compared to those of similar materials produced using the traditional technology. Last of all, these developments are viewed as promising novel approaches that can open new perspectives for the application at larger scale of PLA–ZnO nanocomposites characterized by multifunctional end-use properties (UV protection, antibacterial, gas-barrier, etc.).

## 2. Results and Discussion

It is important to recognize that, for the easier analysis of the results discussed hereinafter, it will be helpfully to consider the information given in the experimental Section 3.2, Section 3.3 and Section 3.4 regarding the two techniques considered in this study to produce PLA–ZnO nanocomposite films:

Method (A) The MB method (in one-step), consisting of the direct mixing of pristine PLA (granules) with highly filled MBs and the extrusion of nanocomposite films using a single screw extruder.

Method (B) For the sake of comparison, the current technique, involving realization of PLA–ZnO nanocomposites (granules) by melt-compounding in twin-screw extruders, followed in the second step, by the extrusion of films using single screw extruders.

As it is mentioned in the experimental section, the PLA–ZnO MBs and the nanocomposites used for comparison as granules were produced using Leistritz modular twin-screw extruders (ZSE 18 HP-40D (screw diameter (D) = 18 mm, L (length)/D ratio = 40) and a specific profile of melt-compounding temperatures. The composition and codification of MB samples produced by melt-compounding are shown in Table 1**.** Function of ZnO amount (i.e., 10%, 20%, 30%, and 40 wt.%), the MBs are abbreviated respectively as MB10, MB20, MB30 and MB40. They are characterized and tested to produce nanocomposite films, after the previous dilution with the neat PLA (Figure 1). Deliberately, the MB40 will be mainly concerned in this study because it requires higher dilution with virgin PLA. It is noteworthy also mentioning that in the case of utilization of MBs, to arrive to desired loading of ZnO and the same content thermal stabilizer in the final products (0.3% U626), the additive was added in the phase of production of films. The MB approach (A) was compared to the current technology (B) to produce PLA–ZnO nanocomposite films. Still, for more clarity, the composition of nanocomposite samples produced as granules and used for comparison (method B) is shown in Table 2. On the other hand, the neat PLA was used in some comparative tests as a kind of reference.

### 2.1. Preliminary Considerations Regarding the Production and Properties of PLA–ZnO MBs

It is worth recalling that PLA is very sensitive to temperature, shear, and hydrolysis during melt-compounding and subsequent processing. Besides, in many cases the PLA mixing with additives and nanofillers (such as ZnO) is followed by the important drop of molecular weights, together with the loss of thermal, rheological, and mechanical properties. However, in the specific case of PLA–ZnO nanocomposites, the degradation of PLA during melt-compounding and processing at a high temperature can be even more advanced due to the presence of ZnO nanofiller [26]. To counterbalance this undesirable effect, the utilization of ZnO NPs surface-treated with silanes (such as triethoxycaprylylsilane) was considered an effective way to limit the decrease of PLA molecular weights with respect to untreated ZnO. Besides, it is essentially to reduce the residence time of PLA at high temperature in the presence of ZnO and this could be possible by utilization of MBs. Accordingly, one key question is connected to the practical possibility to produce highly filled PLA–ZnO MBs. Somewhat surprisingly by considering the high effectiveness of this nanofiller in inducing the degradation of polyester chains, using specifically surface treated NPs, the production by melt-compounding of MBs of PLA filled with up to 40% ZnO using twin-screw extruders is experimentally confirmed (please also consider elsewhere (Section 2.3) the images of MB(40) as granules and that of nanocomposite films produced using the MB method). Selected results obtained following the characterization of PLA–ZnO MBs are discussed below.

#### 2.1.1. Thermal Analyses on MBs

To examine the effects of ZnO addition on the thermal stability of PLA, thermogravimetric analyses (TGA) on MBs have been performed under air (the TG and d-TG curves are shown respectively in Figure 2a,b). First, it is worth pointing out that the nanofiller has a good thermal stability (Figure 2a), thus at the exception of some traces of water, TGA evidenced only a weight loss of about 2.5% up to the temperature of 500 °C, which is mainly ascribed to the degradation of alkyl part of triethoxycaprylylsilane used for the ZnO surface-treatment. As reported elsewhere [26], the addition of ZnO nanofiller (untreated or surface treated) is generally expected to lead to the decrease (more or less) of PLA thermal stability. This is ascribed to the presence of oligomers and polyester chains of low molecular weights. For easier interpretation, Table 3 shows the values of the temperatures corresponding to 5% weight loss and to the maximum rate of degradation (respectively, T_5%_ and T_D_—from d-TG). As a remark, T_5%_ is often considered as the initial decomposition temperature. The thermal stability of MBs is lowered by comparing to the neat PLA (T_5%_ in the range 270–280 °C compared 326 °C, respectively), which is typically expected. However, it is noteworthy mentioning that in the case of MBs, T_5%_ is remaining much higher than the max. temperature of PLA processing (about 220 °C). Still at different ZnO loadings in MBs, somewhat surprisingly, there are only few differences between samples (i.e., T_5%_ is between 270 °C and 278 °C, while T_D_ is found in the range 300–315 °C).

In relation to the thermal parameters as evaluated by DSC analysis on MBs as granules after their drying (Figure 3), it comes out that there are seen minor differences, mostly ascribed to the percentage and effects of ZnO. Obviously, from the DSC results obtained in the first heating scan it is evident that all samples are characterized by important melting enthalpies, suggesting a high crystallinity achieved by drying (degree of crystallinity >35%), while the melting temperature (T_m_) values are in the range 166–168 °C. By considering the important degree of crystallization, a supplementary post-crystallization process will not be required for MBs to avoid the sticking of granules on the screws of extruders in the feeding zone. Still, regarding the information obtained in the second DSC heating (after cooling from the molten state by 10 °C/min), from the quantification of data (Table 4) it is assessed that even adding 10–40% ZnO into PLA, the filler has not some spectacular effects in increasing the speed of crystallization of PLA matrix, the degree of crystallinity of PLA being low (3–5%). Concerning the other thermal events (i.e., glass transition temperature (T_g_), cold crystallization temperature (T_cc_), melting temperature (T_m_)), the minor diminishing of values by increasing ZnO loading can be mainly ascribed to the presence of products of low molecular weights, such as lactide/lactic acid oligomers with (slight) plasticizing role, while it is not excluded the influence of other factors (NB: such as connected to the treatment of nanofiller with a silane containing lipophilic caprylyl groups).

#### 2.1.2. Rheological Investigations on PLA–ZnO MBs

MFI measurements were carried out to have the first information about the rheology of MBs for melt-blending and processing by extrusion. From the values of MFI (Figure 4), it is evident that the addition of high amounts of ZnO strongly affects the PLA melt flowing/rheology, a higher MFI being noticeable for MBs by comparing to the neat PLA (MFI in the range 14–23 g/10 min compared to 4 g/10 min, respectively). The important increase of MFI/fluidity observed even at low volume fraction of nanofiller (~2.4% vol) by addition of 10 wt.% ZnO into PLA (MFI of 22.8 g/10 min for MB10), can be once more explained by the presence of PLA chains of low molecular weight, while as mentioned before, it is not excluded an additional lubricating effect due to the presence of alkyl-silane on the surface of ZnO NPs. Concerning the highly filled MBs (30–40% ZnO), the diminishing of MFI values to about 13 g/10 min can be interpreted as an increase in melt viscosity due to high loading with nanofiller. Still, it is assumed that MBs characterized by higher fluidity (NB: compared to the pristine PLA) will present a better aptitude for melt-blending and mixing with PLA of medium or high viscosity, respectively, grades for spinning or film extrusion. In forthcoming research, additional characterizations will be of interest to have more information about the evolution of PLA molecular weights in MBs and in relation to the rheological behavior under specific conditions (e.g., using rotational rheometers).

### 2.2. Comparative Testing of PLA–ZnO MBs by Extrusion to Produce Nanocomposite Films

Since, in the previous studies, the anti-UV and antibacterial properties were confirmed respectively, at 1% and 3% ZnO loading [26,30], only the nanocomposite films loaded with up to 3% ZnO will be hereafter concerned.

#### 2.2.1. PLA–ZnO Films Produced Using Alternative Methods

As suggested in Figure 5, two methods for producing films (100–400 µm thickness) by extrusion are compared in this study (see also the experimental section): (A) the direct mixing of pristine PLA with highly filled MB40 (in the presence of 0.3% U626) and the extrusion of films (new method, in one step); (B) The traditional technique, in two-steps, including realization of PLA–ZnO nanocomposites as granules (by melt-compounding using twin-screw extruders), followed by the extrusion of films via a single screw extruder.

As mentioned before, deliberately, the MB with the highest percentage of ZnO (MB40) was selected for discussion because it needs higher dilution with virgin PLA. To have similar amount of thermal stabilizer (0.3% U626), the additive vas added in the phase of dry mixing with the granules of PLA, before the extrusion of films. As exemplified in Figure 5, to produce PLA–1% ZnO nanocomposite films, in the case of traditional procedure in two-steps (B), all amount of PLA (98.7%) is in contact with the ZnO (1%) for long time at high temperature and shear. On the other hand, using the MB40 (method A) and the direct mixing with the neat PLA (0.3% U626A added as thermal stabilizer) to produce comparable nanocomposite films, 97.2 wt.% PLA will have much shorter residence time at high temperature in the presence of ZnO, noteworthy to state, only in the single screw extruder used to produce films. In fact, only 1.5 wt.% PLA (from MB) has longer thermal history and residence time. Accordingly, it is expected to significantly limit the degradation of polyester chains using MBs, with benefits in terms of better processing, increasing in melt-strength and enhanced properties. Indeed, the different parameters recorded during extrusion of films (Table 5) are assessing for an improved processing typically observed in the case of PLA–MB40 blends (higher torque and pressure). This enhancement is reasonably ascribed to the higher viscosity in the molten state due to the lower degradation/better preserving of PLA molecular weights.

#### 2.2.2. Characterization of Nanocomposite Films Produced Using Alternative Methods

Henceforth, the nanocomposites produced by melt compounding in twin-screw extruders and processed as films (for the sake of comparison) will be abbreviated as PLA–x% ZnO, while the films produced using MBs will be noted as PLA–x% ZnO_(MB)_ (in both cases, x wt.% being the percentage of ZnO).

(a)Viscosimetric analyses: The modification of PLA molecular weights in nanocomposite films was first studied by performing measurements of relative viscosity in solution (chloroform as solvent). Undeniably, the relative viscosity (Figure 6) of polymer samples from films produced via MB route is much higher than those of similar films obtained using traditional procedure (B): respectively, values in the range 3.1–3.4 compared to 2.2–2.4. These results allow concluding that the molecular weights of PLA chains are better preserved using MBs, thus they can explain the improvements in terms of processing and performances (vide infra). It is also noteworthy mentioning that in additional tests, after a very carefully separation of ZnO nanofiller, the films were characterized by size exclusion chromatography (SEC) to have more exact information about the molecular weights of the polyester chains in these nanocomposites (please also consider Appendix A). The SEC results were found in relatively good agreement with those of relative viscosity, the molecular weights of PLA in nanocomposite films being significantly higher using MBs (approach A) than in the case of traditional method (B) (e.g., at 3% ZnO loading in nanocomposite films, PLA has a M_n(PS)_ of 92600 and 54800, respectively).

(b)Thermal properties of films: TG measurements on films produced according to both methods have been compared using non-isothermal tests (Table 6). Taking as parameter the onset of thermal degradation (T_5%_), in all cases, addition of ZnO as premix (MB) in PLA leads to nanocomposites that show enhanced thermal stability. Such improvements, i.e., the increasing of T_5%_ by 10 °C to 30 °C, can be readily ascribed to the lower thermal degradation of PLA and better preservation of molecular weights in nanocomposites as films. Moreover, these enhancements are reasonably assigned to the important decrease of residence time at high temperature of PLA in the presence of ZnO NPs.

Differential Scanning Calorimetry (DSC): Regarding the DSC results obtained by characterization of films (comparative DSC traces shown in Figure 7, DSC data summarized in Table 7), it comes out that there are not substantial changes for the values of T_g_ (glass transition temperature) which can be connected to the differences of ZnO loading or to the techniques of production. Still, as it was previously reported elsewhere [34], it is reconfirmed that this specific treated nanofiller is not of high effectiveness in promoting properties of crystallization for PLA, the films having a low degree of crystallinity. Besides, this parameter can be somewhat influenced by the inherent drawing process during the cooling of films on the conveyer belt. On the other hand, by considering the nanocomposites with similar amount of ZnO, using MBs it is seen a short delay in PLA crystallization. The peak of cold crystallization (T_cc_) is detected slightly at higher temperature, while the degree of crystallinity is somewhat lower, and this is reasonably ascribed to the difficulty of PLA of high molecular weight to crystallize. The presence of two melting peaks (more evident for the neat PLA and PLA-MB blends) or of shoulders on the endotherms of melting, is an indication of the presence of different kind of crystals (such as α′- and α crystals [38]) with different lamellae thickness, while is not excluded the influence of other factors, such as connected to the extrusion and drawing process.

(c)Morphology of nanocomposite films: Figure 8a–i shows selected TEM images of ZnO nanofiller and of nanocomposite films (with 1% and 2% ZnO NPs) produced following the different methods. The primary NPs are characterized by a rod-like morphology, with diameters in the range 10–30 nm and a length up to about 100 nm. Regarding the morphology of nanocomposites produced using MBs, due to the high let down ratio (MB/neat PLA) and experimental conditions (short time of mixing in a laboratory single screw extruder, at moderate screw speed/shear), can be some questions in relation to the quality of ZnO distribution/dispersion when the highly filled MB (40% ZnO) is mixed with pristine PLA. Noticeably, as it is suggested by the micrographs shown in Figure 8b,d, it is obvious from the TEM images of nanocomposite films obtained using MB that the ZnO NPs are quite well-distributed through PLA matrix. Still, at higher magnification (Figure 8c,e), the presence of isolated NPs and clusters composed of few ZnO rods appears to be more common. However, a better dispersive mixing will certainly be obtained in the case of industrial extrusion equipment with enhanced melt-blending capability and at increased shearing [39,40]. On the other hand, it is noteworthy reminding that the surface coating of ZnO by alkyl-silane is providing finer distribution and dispersion of NPs through PLA matrix [26,30], and these expectations are confirmed once more for the films produced using nanocomposites (as granules) following the conventional method B (Figure 8f–i).

(d)Mechanical properties (Table 8): The mechanical characterizations of films obtained using different techniques are revealing only minor differences regarding the tensile strength characteristics, i.e., values in the range 57–60 MPa, slightly better using MB. As mention, Appendix A shows typical tensile stress—strain curves of PLA–ZnO nanocomposite films produced using both methods. Interestingly, even that the PLA molecular weights were diminished using the method B, the molecular parameters are remaining high enough to allow good mechanical properties. However, the improvements in rigidity (Young’s modulus) are more evident using MBs, enhancement in quite good correlation with the amount of nanofiller (ZnO). Still, in all cases, these films (MB method, A) have better elongation at break (ε_b_) with respect to the films produced using the conventional approach (B).

### 2.3. Specific End-Use Properties and Current Prospects in the Field of PLA–ZnO MBs

It is noteworthy restating that the PLA–ZnO nanocomposites have been already tested in the production of films and fibers as new (bio)sourced and biodegradable materials characterized by multifunctional properties (anti-UV, antibacterial, barrier, etc.) [26,30]. Henceforth it is compared the UV protection of nanocomposite films produced using the different methods. ZnO acts as a transparent broad-band UVA and UVB absorber, thus the polymeric films can be characterized by excellent UV protection properties, with high levels of transparency in the case of thin films. On the other hand, it is known that the neat PLA do not offer any UV protection. Therefore, for the utilization of PLA in applications, such as dairy products, are required specific additives to block UV light transmission and to prevent the damage of light sensitive packaged foods, with beneficial effects on the retention of taste and appearance, extension of shelf life and improvement of product quality [6]. Figure 9a,b shows the comparative UV-vis transmittance spectra of PLA and PLA–ZnO nanocomposites (films of about 100-micron thickness) with different loadings of ZnO obtained using the alternative methods considered in this work. The films of PLA have an absorption band with a maximum at about 275 nm in UV−region of spectra, which can be attributed to the phenolic function of the U626 used as thermal stabilizer [31]. From Figure 9a, it comes out that the addition of ZnO as MB leads to nanocomposite films showing a good UV-shielding effect, while the light transmittance in the visible range of the spectra is significantly influenced by ZnO loading. The optical losses and important reduction of transmittance for the films having higher nanofiller amount, are probably due to the scattering of visible light, absorption, and reflection effects of ZnO NPs and inherent ZnO clusters. These results are comparable to those on films (Figure 9b) obtained using the traditional production method (B), which could be theoretically credited with better morphology/nanofiller dispersion (NB: assumption connected to the specific technology of production, see the experimental part). Still, the differences in the light transmittance in the visible range are evidently affected by the loading of ZnO and by the thickness of films.

Regarding the most recent developments to produce MBs for the extrusion of films and fibre spinning, additional forthcoming studies have been performed to improve their characteristics, by the direct addition of thermal stabilizers (U626) and of chain extenders (CE) in the phase of production of PLA–ZnO MBs. Indeed, the addition of multi-functional CE from the category of epoxy functional styrene-acrylate oligomers [41,42] into PLA–ZnO compounds is seen as an attractive low-cost method to revert the molecular weight degradation, and finally, a way to produce more performant nanocomposites [34]. Also, for realization of films by extrusion, in the last experiments a flat film take-off unit working with nip rolls allowing controlled temperature was successfully tested as downstream equipment.

Furthermore, it is assumed that the percentage of nanofiller (ZnO) from MBs will need carefully optimization considering the requirements of specific processing and final application, aspects not concerned enough in this study. The utilization of MBs is considered a novel technological progress to produce more competitive PLA–ZnO nanocomposites (images of MB(40) (granules) and of nanocomposite films produced using the MB technique are shown in Figure 10a–c). Finally, it is noteworthy mentioning that the MB method offers to processors more flexibility in the choice of ZnO loading function of application, the possibility to use varied techniques of processing (extrusion, spinning, injection molding), cumulated benefits such as cost savings and technical performances, cleaner facilities, no health, and safety/pollution risks connected to the manipulation of nanofillers, and so on. PLA nanocomposites characterized by multifunctional end-use properties are of a high interest as biomaterials to produce films, textile fibers, injection molded items.

## 3. Materials and Methods

### 3.1. Materials

Poly(L,L-lactide)—hereafter called PLA, supplier NatureWorks LLC (Minnetonka, MN, USA), was a grade designed for the extrusion of films (4032D) with M_n(PS)_ = 133,000, dispersity, M_w_/M_n_ = 1.94 (M_w_ and M_n_, being respectively, the weight- and number-average molar mass expressed in polystyrene equivalent), whereas according to the producer the other characteristics are as follows: D-isomer = 1.4%; relative viscosity = 3.94; residual monomer = 0.14%.

Silane-treated ZnO nanofiller was supplied by Umicore Zinc Chemicals (Angleur, Belgium) as Zano 20 Plus (surface treated with triethoxycaprylylsilane, ZnO content above 96%). According to our TEM (Transmission Electron Microscopy) investigations, these nanoparticles are characterized by a rod-like morphology, having diameters of 10–30 nm and a maximum length of about 100 nm.

Ultranox 626A (Bis (2,4-di-t-butylphenyl) pentraerythritol diphosphite) supplied by Brenntag NV (abbreviated as U626), was selected as thermal stabilizer, and used at a preferred percentage of 0.3% in all PLA compositions designed for the production by extrusion of films. Throughout this contribution, all percentages are given as weight percent (wt.%).

### 3.2. Preparation of PLA–ZnO Masterbatches

Before melt-compounding, PLA and ZnO nanofiller were dried at 70 °C overnight using a drying oven with recirculating hot air. Then, PLA MBs with up to 40% ZnO were prepared by melt-compounding using a twin-screw extruder Leistritz ZSE 18 HP-40D (screw diameter (D) = 18 mm, L (length)/D ratio = 40) and the following processing conditions: the speed of screws = 150 rpm, temperatures of extrusion on the heating zones: Z1= 185 °C; Z2= 210 °C; Z3= 210 °C, Z4-Z7 = 200 °C; while the temperature of the extrusion die was kept at 190–200 °C. Two separate gravimetric feeders have been used for the dosing of PLA and of ZnO nanofiller. The loading of nanofiller in PLA–ZnO MBs was increased by steps of 10% (10%, 20%, 30%, and 40%).

### 3.3. Preparation of PLA–ZnO Nanocomposites Following the Traditional Method

For the sake of comparison, PLA–ZnO nanocomposites with 1%, 2% and 3% ZnO have been prepared by melt-compounding using the same twin-screw extruder equipment. According to the conventional method, firstly the PLA granules were dry-mixed in a Rondol turbo-mixer with the nanofiller and 0.3% U626 (thermal stabilizer), step followed by the direct feeding of the blend in the twin-screw extruder for melt-compounding. Conditions of melt-compounding: the speed of the screws = 150 rpm; the temperatures of extrusion on the heating zones were adapted to the rheological characteristics of PLA blend (Z1 = 185 °C; Z2 = 195 °C; Z3 = 205 °C; Z4–Z6 = 200 °C; Z7 = 195 °C).

### 3.4. Comparative Evaluation of PLA–ZnO MBs and Nanocomposites in Production of Films

Practically, the drying of granules (PLA, MBs, PLA–ZnO nanocomposites) at 70 °C overnight was followed by realization of films using a Brabender laboratory single screw extruder (D = 19 mm, L/D = 25). For the direct comparison of two methods, the dry-mixed blends of MB (40% ZnO) and pristine PLA (with 0.3% U626) in method A or the pellets of nanocomposites (method B), were extruded as films of different thickness (0.1–0.4 mm) using a rectangular extrusion die head (ribbon die of 50 mm wide, gap of 0.5 mm). Temperatures on the heating zones of single screw extruder: Z1 = 200 °C; Z2 = 205 °C; Z3 = 210 °C; temperature of the extrusion head: 215 °C. As, downstream equipment for drawing and cooling, a silicone-coated conveyor belt was used to take the films and to obtain the required film thickness.

### 3.5. Characterization

#### 3.5.1. Relative Viscosity Measurements

The relative viscosities in solution were determined at the temperature of 25 °C after dissolving the polymer samples (films) in chloroform (1 g polymer/100 mL chloroform) and using an automated Ubbelohde viscosimeter (SCHOTT^®^ Geräte, capillary type R 53113/IC). Due to the presence of nanofiller in PLA, for the protection of GPC/SEC columns in the first experiments, it was preferred to perform analyses of viscosity in solution, rather than the more conventionally size exclusion chromatography (SEC) analysis that was considered later (details as supplementary information).

#### 3.5.2. Thermogravimetric Analysis (TGA)

TGA were performed by using a TGA Q50 (TA Instruments, New Castle, DE, USA) at a heating ramp of 20 °C/min under air flow, from room temperature up to 600 °C (platinum pan, 60 cm^3^/min air flow rate).

#### 3.5.3. Differential Scanning Calorimetry (DSC)

DSC measurements were performed by using a DSC Q200 from TA Instruments (New Castle, DE, USA) under nitrogen flow. The procedure was as follows: first heating scan at 10 °C/min from 0 °C up to 200 °C, isotherm at this temperature for 2 min, then cooling down at 10 °C/min to −20 °C and finally, second heating scan from −20 to 200 °C at 10 °C/min. The first DSC scan was used in some cases to erase the prior thermal history of the samples. The events of interest, i.e., the glass transition temperature (T_g_), cold crystallization temperature (T_cc_), enthalpy of cold crystallization (ΔH_cc_), melting temperature (T_m_) and melting enthalpy (ΔH_m_) were typically followed in DSC heating scans. Noteworthy, for all samples, only the fraction of PLA was considered. The degree of crystallinity was determined by subtracting ΔH_cc_ from ΔH_m_ and by considering a melting enthalpy of 93 J/g for 100% crystalline PLA.

#### 3.5.4. Rheological Measurements

The melt flow index (MFI) of selected samples was determined using an automatic Davenport MFI 10 melt flow indexer. The measurements were performed at the temperature of 190 °C following the procedure described in ASTM D1238 (standard die 2.095 × 8 mm and using a 2.16 kg load).

#### 3.5.5. Mechanical Testing

Tensile testing measurements were performed on rectangular strips (64 × 10 × ~0.4 mm^3^, cut from the extruded films) according to ASTM D 638-02a and using a Lloyd LR 10K tensile bench at a speed rate of 1 mm/min, with 25.4 mm distance between grips. All mechanical tests were carried out by using specimens previously conditioned for at least 48 h at 20 ± 2 °C under a relative humidity of 50 ± 3% and the values were averaged out over minimum five measurements. The NEXYGEN™ MT Materials Test and Data Analysis Software was used to configure the tests, analyze, and report the results. The nominal strain was determined as the change in grip separation relative to the original grip separation expressed as a percent. The values of Young’s modulus were obtained directly via software and validated for accuracy considering two methods: 1. the secant modulus at low deformation (between 0.05% and 0.25% strain) and 2. the initial tangential modulus (values reported as Young’s modulus).

#### 3.5.6. Transmission Electron Microscopy (TEM)

Transmission electron micrographs on selected samples were obtained with a Philips CM200 apparatus using an accelerator voltage of up to 120 kV. The samples (70–80 nm thick) were prepared with a Leica Ultracut UCT ultracryomicrotome by cutting them at −100 °C. Reported microphotographs represent typical morphologies as observed at, at least, three different locations.

#### 3.5.7. UV Absorption Properties

Ultraviolet and visible (UV-Vis) absorption spectroscopy analyses were realized on films of selected thickness using an UV-Vis NIR Cary Series (Agilent Technologies, Santa Clara, CA, USA) spectrophotometer. The blank reference was air. Transmittance spectra were recorded in the 200–800 nm wavelength range by considering a minimum of three measurements.

## 4. Conclusions

PLA–ZnO nanocomposites are of a high concern as new biomaterials characterized by multifunctional end-use properties (anti-UV, antibacterial, gas barrier, etc.). Since the addition of untreated ZnO NPs into PLA at high melt-processing temperature leads to the severe degradation of polyester macromolecular chains, the surface treatment of ZnO is a prerequisite to limit this undesired effect. The study highlights the beneficial effects of premixing ZnO with PLA under given conditions and the use as masterbatch (MB) as alternative manufacturing technique. Two methods are compared to produce nanocomposite films: the MB approach (A) and the current method in two-steps (B), i.e., production of PLA–ZnO nanocomposites (as granules) by melt-compounding in twin-screw extruders, followed by the extrusion of films (single screw extruder).

First, the study assesses that the production of highly filled MBs with silane treated (triethoxycaprylylsilane) ZnO NPs (10%–40 wt.%) using twin screw-extruders is feasible. The MBs have been characterized point of view thermal and rheological properties. Then, PLA–(1–3%) ZnO nanocomposite films were produced to compare the two alternative techniques, MB and current method, both approaches being developed following original studies. Better processing (rheology), higher molecular weights, enhanced thermal stability (T_5%_ is raised with 10 to 30 °C), (moderate) increase of mechanical properties, good nanofiller distribution, powerful UV protection, are characteristic features ascribed to the films produced using MBs. The MB technique allows us to shorten the residence time at high temperature of PLA in the presence of ZnO. Thus, it is highly beneficial for the preservation of PLA molecular weights. Furthermore, utilization of PLA–ZnO MBs is seen as an important opportunity to lead to more competitive products, close to the industrial processing conditions. These developments are viewed as a promising novel and more flexible method (for users) that can open new perspectives for the larger utilization of multifunctional PLA–ZnO nanocomposites for various applications (packaging films, textile fibers, injection molded items, etc.).

## Figures and Tables

**Figure 1 molecules-26-02043-f001:**
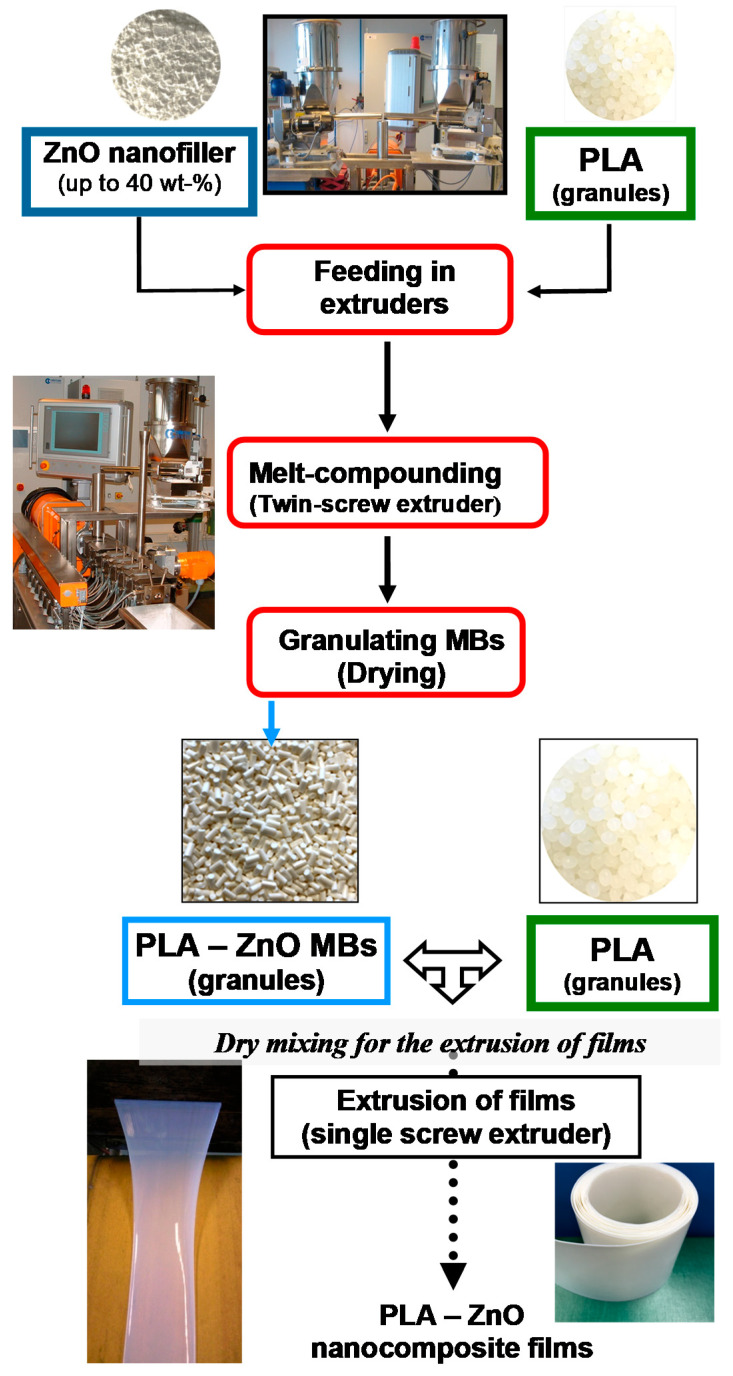
Principal steps in realization of poly(lactic acid) (PLA)–ZnO masterbatches (MBs) and the testing to produce nanocomposite films.

**Figure 2 molecules-26-02043-f002:**
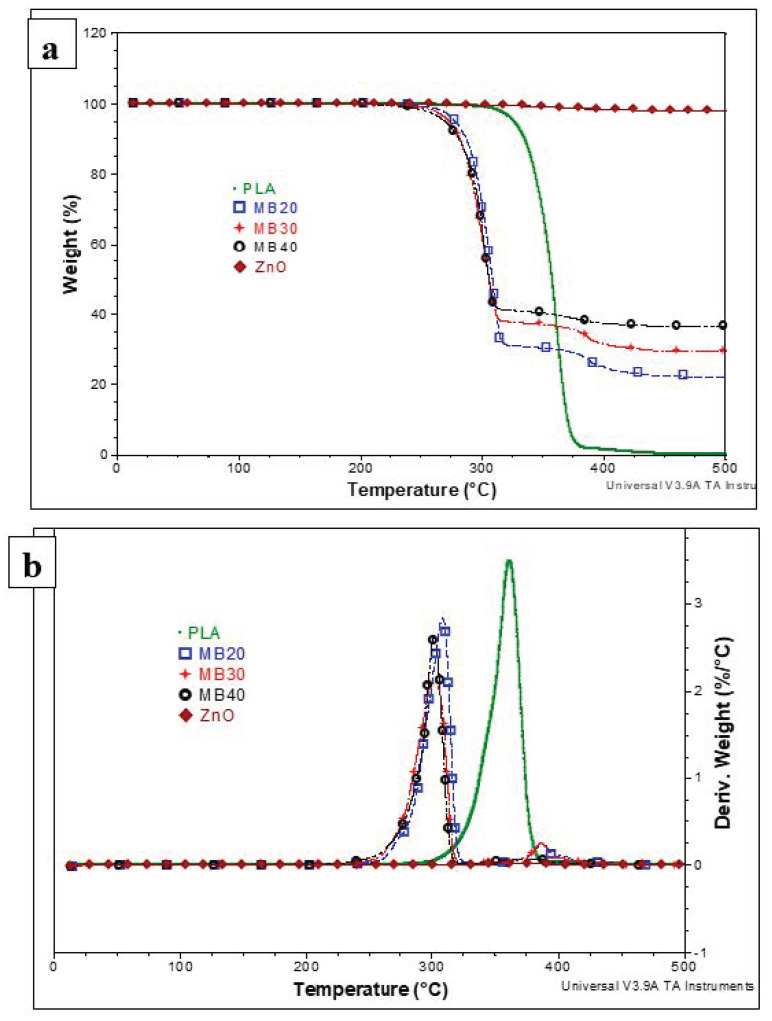
Comparative thermogravimetric (TG) (**a**) and derivative (d-TG) (**b**) traces of ZnO nanofiller, neat PLA and of highly filled PLA–ZnO MBs (under air, 20 °C/min).

**Figure 3 molecules-26-02043-f003:**
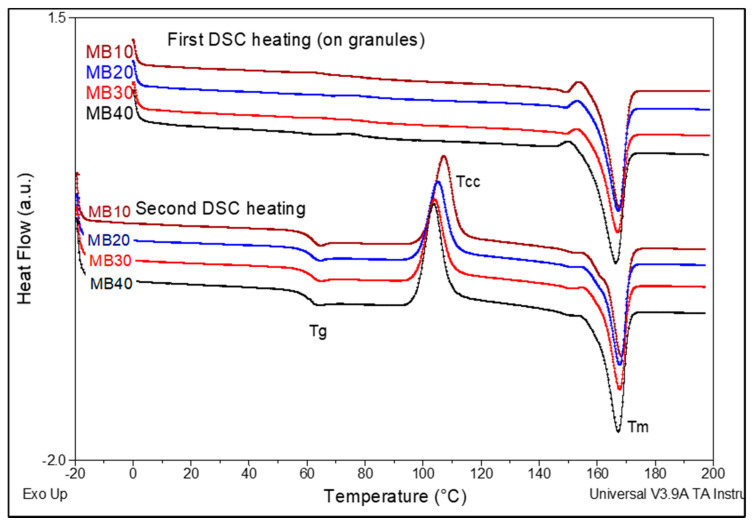
Comparative Differential Scanning Calorimetry (DSC) thermograms of PLA–(10–40)% ZnO MBs (first and second DSC heating, 10 °C/min).

**Figure 4 molecules-26-02043-f004:**
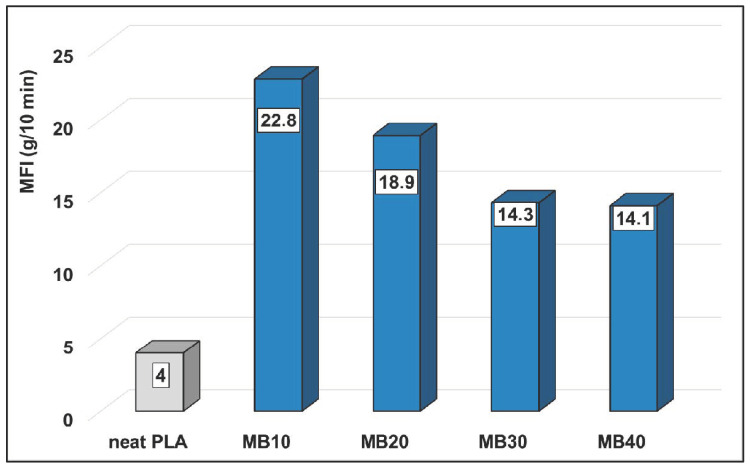
Comparative MFI values (190 °C, 2.16 kg) of neat PLA and PLA–ZnO MBs.

**Figure 5 molecules-26-02043-f005:**
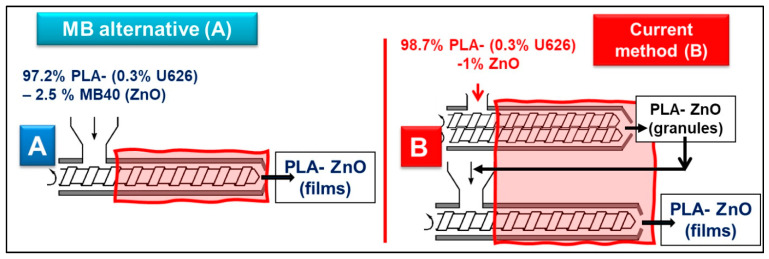
Alternative techniques to produce PLA–1% ZnO nanocomposite films: (**A**)—MB option; (**B**)—current method. The zones in red color are suggesting the residence time at high temperature of PLA in the presence of ZnO.

**Figure 6 molecules-26-02043-f006:**
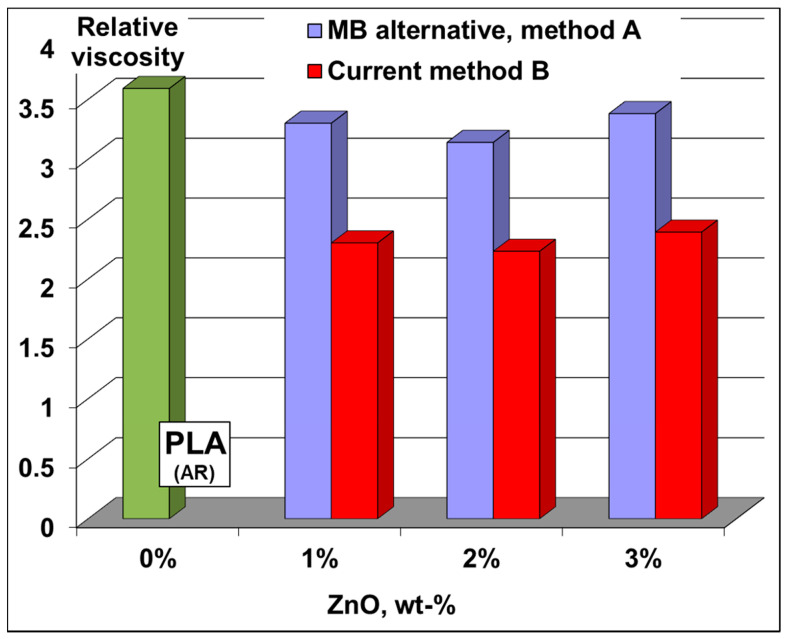
Relative viscosities (in chloroform) of neat PLA and PLA nanocomposite films obtained using the MB approach (**A**) versus the traditional method in two-steps (**B**).

**Figure 7 molecules-26-02043-f007:**
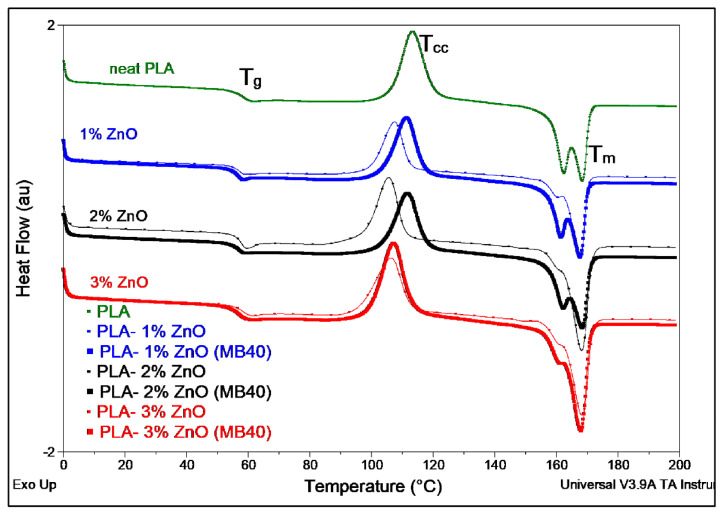
DSC traces obtained in the first heating scan (10 °C/min) on PLA and PLA–ZnO nanocomposite films.

**Figure 8 molecules-26-02043-f008:**
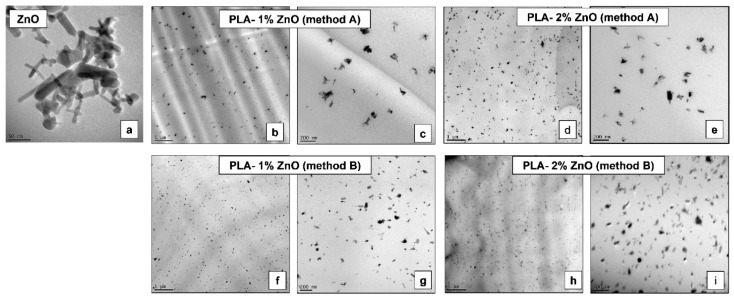
(**a**–**i**). TEM pictures to illustrate the morphology of ZnO nanofiller (**a**), PLA–1% and 2% ZnO nanocomposite films obtained using the MB approach (**b**–**e**) and the traditional technique (**f**–**i**).

**Figure 9 molecules-26-02043-f009:**
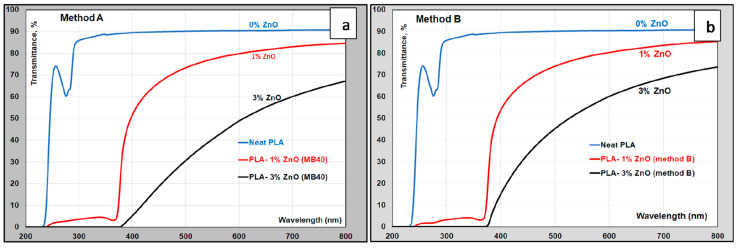
UV-vis transmittance spectra of PLA and PLA–(1–3)-% ZnO nanocomposite films (thickness ~100 µm) produced using: (**a**) the MB approach; (**b**) PLA–ZnO nanocomposites as granules (method B).

**Figure 10 molecules-26-02043-f010:**
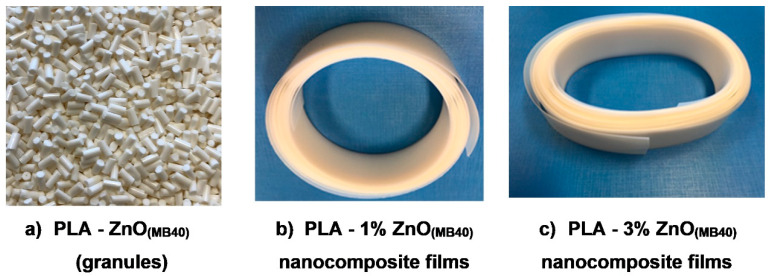
Aspect of MB(40) as granules (**a**) and of nanocomposite films (~400 µm thickness) produced using the MB technique: (**b**) PLA–1% ZnO_(MB40)_ and (**c**) PLA–3% ZnO_(MB40)_ nanocomposite films.

**Table 1 molecules-26-02043-t001:** Composition (wt.%) and codification of MBs obtained by melt compounding using twin-screw extruders.

Sample Code	PLA, %	ZnO, %	U626, %
MB10	90	10	*
MB20	80	20	*
MB30	70	30	*
MB40	60	40	*

* In the case of MBs, to keep a similar amount of thermal stabilizer in final products (0.3% U626) the additive was added in the phase of production of films.

**Table 2 molecules-26-02043-t002:** Composition (wt.%) of PLA–ZnO nanocomposites (granules) obtained by melt compounding using twin-screw extruders for comparative testing to produce films (method B).

Sample Code	PLA, %	ZnO, %	U626, %
PLA–1% ZnO	98.7	1	0.3
PLA–2% ZnO	97.7	2	0.3
PLA–3% ZnO	96.7	3	0.3

**Table 3 molecules-26-02043-t003:** Selected TGA results of PLA and PLA–(10–40%) ZnO MBs (under air flow, 20°C/min).

Sample	Temperature at 5% Weight Loss, °C	Temperature at Max. Rate of Degradation (from d-TG), °C
PLA	326	361
MB10	277	314
MB20	278	308
MB30	273	301
MB40	270	302

**Table 4 molecules-26-02043-t004:** Comparative DSC data of PLA–ZnO MBs (second DSC heating, 10 °C/min) *.

Sample	T_g_ (°C)	T_cc_ (°C)	ΔH_cc_ (J g^−1^)	T_m_ (°C)	ΔH_m_ (J g^−1^)	χ_c,_ %
PLA	63	113	30.4	169	31.7	1.4
MB10	62	107	31.3	168	34.6	3.5
MB20	62	105	28.2	168	31.0	3.0
MB30	62	104	28.1	168	31.4	3.5
MB40	61	104	34.2	167	38.7	4.8

* Abbreviations: glass transition temperature (T_g_), cold crystallization temperature (T_cc_), enthalpy of cold crystallization (ΔH_cc_), melting temperature (T_m_), melting enthalpy (ΔH_m_), degree of crystallization (χ_c,_).

**Table 5 molecules-26-02043-t005:** Comparative parameters recorded during extrusion of PLA–1% ZnO nanocomposite films.

Parameter	MB Technique (A)	Current Method (B)
Speed of the screw, rpm	12	12
Torque, Nm	110	20
Pressure of extrusion, bars	50	3

**Table 6 molecules-26-02043-t006:** Onset of thermal degradation (T_5%_) and decomposition temperature (T_D_) as determined by TGA of PLA–ZnO nanocomposite films (under air flow, 20 °C/min).

Sample (%, by Weight)	Method of Production	Temperature at 5% Weight Loss (T_5%_), °C	Temperature at Max Rate of Degradation (T_D_), °C
PLA–1% ZnO_(MB40)_	A	297	340
PLA–1% ZnO	B	285	343
PLA–2% ZnO_(MB40)_	A	300	342
PLA–2% ZnO	B	272	320
PLA–3% ZnO_(MB40)_	A	294	335
PLA–3% ZnO	B	274	321

**Table 7 molecules-26-02043-t007:** DSC on PLA and PLA–ZnO nanocomposite films obtained using different methods (first DSC heating scan, by 10 °C/min) *.

Sample (%, by Weight)	T_g_ (°C)	T_cc_ (°C)	ΔH_c_ (J g^−1^)	T_m_ ** (°C)	ΔH_m_ (J g^−1^)	χ_c,_ %
PLA	59	113	34.4	162; 168	38.8	4.8
PLA–1% ZnO_(MB40)_	57	111	29.9	162; 168	35.0	5.5
PLA–1% ZnO	57	107	22.3	161; 168	29.0	7.2
PLA–2% ZnO_(MB40)_	57	112	31.5	162; 168	37.7	6.7
PLA–2% ZnO	58	106	32.6	160; 168	41.1	9.1
PLA–3% ZnO_(MB40)_	57	107	37.6	161; 168	43.2	6.0
PLA–3% ZnO	58	106	31.4	160; 168	38.2	7.3

* Abbreviations: glass transition temperature (T_g_), cold crystallization temperature (T_cc_), enthalpy of cold crystallization (ΔH_cc_), melting temperature (T_m_), melting enthalpy (ΔH_m_), degree of crystallization (χ_c_). ** presence of two melting peaks or of a shoulder and a main melting peak, see Figure 7.

**Table 8 molecules-26-02043-t008:** Mechanical properties of PLA–ZnO nanocomposite films obtained using different approaches (v = 1 mm/min, films of ~0.4 mm thickness).

Entry	Sample (%, by Weight)	Max. Tensile Strength, MPa	Young’s Modulus, MPa	Nominal Strain at Break, %
1	PLA–1% ZnO_(MB40)_	58 (±2)	2600 (±150)	8.1 (±0.8)
2	PLA–2% ZnO_(MB40)_	60 (±3)	2700 (±150)	8.2 (±1.8)
3	PLA–3% ZnO_(MB40)_	58 (±1)	2750 (±50)	8.6 (±1.6)
4	PLA–1% ZnO	59 (±2)	2600 (±150)	5.5 (±0.6)
5	PLA–2% ZnO	57 (±4)	2600 (±200)	3.5 (±0.2)
6	PLA–3% ZnO	57 (±2)	2650 (±100)	3.9 (±0.3)

## Data Availability

Pieces of information connected to this study have been presented as poster communication and are available as supporting material on the ResearchGate website page of M.M. at the following address [https://www.researchgate.net/publication/273137029_Adding_value_in_production_of_PLA_-ZnO_nanocomposites_through_novel_techniques accessed on 7 March 2021]. Data citation: Murariu, M.; Dechief, A.L.; Benali, S.; Dubois, P. et al., Adding value in production of PLA–ZnO nanocomposites through novel techniques, MdC 2015, 10 March 2015, Mons, Belgium.

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
