# Peer review of "Adding Value in Production of Multifunctional Polylactide (PLA)–ZnO Nanocomposite Films through Alternative Manufacturing Methods"

_molecules, 2021, doi:10.3390/molecules26072043_

Round 1

Reviewer 1 Report

The manuscript presents a new method for obtaining PLA - ZnO nanocomposites. To this end, the authors compared the methods already known in previous research investigations and demonstrated that the PLA - ZnO nanocomposites obtained by the method proposed by them are of the same quality as the other methods. The paper is well structured and the authors use appropriate methods for their research

I found that the manuscript can be published after minor revision to the following omissions in it

  1. The following conclusion (in Page 15) is incorrect: “while the light transmittance in the visible range of the spectra is slightly influenced by ZnO loading”.
  2. The spectra presented in Fig. 9 demonstrated significant reduction of transmittance, T in visible spectral range d increase of optical losses, probably due to the scattering from ZnO nanoparticles.
  3. It should be specifiedin the caption which parameters Table 7 presents.
  4. The following expression “UV-vis spectra” in the text and caption of Figure 9 must be replaced with “UV-vis transmittance spectra”

Author Response

Detailed Responses to Reviewers:

Reviewer #1 (R1): “The manuscript presents a new method for obtaining PLA - ZnO nanocomposites. To this end, the authors compared the methods already known in previous research investigations and demonstrated that the PLA - ZnO nanocomposites obtained by the method proposed by them are of the same quality as the other methods. The paper is well structured, and the authors use appropriate methods for their research.

I found that the manuscript can be published after minor revision to the following omissions in it.

  1. The following conclusion (in Page 15) is incorrect: “while the light transmittance in the visible range of the spectra is slightly influenced by ZnO loading”.
  2. The spectra presented in Fig. 9 demonstrated significant reduction of transmittance, T in visible spectral range d increase of optical losses, probably due to the scattering from ZnO nanoparticles.
  3. It should be specified in the caption which parameters Table 7 presents.
  4. The following expression “UV-vis spectra” in the text and caption of Figure 9 must be replaced with “UV-vis transmittance spectra”.”

Authors: Firstly, we would like to thank the Reviewer for the general appreciation of our manuscript and positive comments. Indeed, following the interest for new biomaterials characterized by multifunctional properties, it was first reported by our research group (Biomacromolecules 2011, 12, 5, 1762–1771) the possibility to produce PLA - ZnO nanocomposites via melt-blending technology and using ZnO NPs previously surface‐treated with a specific silane (i.e., triethoxycaprylylsilane) to limit polyester (PLA) degradation. Then, following the contact with potential users, additional developments have been realized because for the industrial application there is a need to find the most adapted techniques to produce more performant products. This study highlights the beneficial effects of MB addition on the processing and properties of PLA - ZnO nanocomposite films as final products. The utilization of MBs can offer more flexibility to the industrial users in relation to the choice of ZnO loadings by considering the final requirements of application (i.e., UV shielding, antibacterial protection, others).

Please find below our point-by-point answers.

R1: “1.            The following conclusion (in Page 15) is incorrect: “while the light transmittance in the visible range of the spectra is slightly influenced by ZnO loading”.

Authors: In totally agreement with the remark of Reviewer we have done the necessary correction, i.e., by replacing “slightly” by “significantly”: “while the light transmittance in the visible range of the spectra is significantly influenced by ZnO loading” Please also consider hereinafter (point 2) the comments in connection to the same subject.

R1: 2.  “The spectra presented in Fig. 9 demonstrated significant reduction of transmittance, T in visible spectral range d increase of optical losses, probably due to the scattering from ZnO nanoparticles.”

Authors: We thank the Reviewer for comment, a good possibility to improve our manuscript. Accordingly, as it is mentioned at the point 1, we are using the term “significant” reduction in transmittance (T) in the revised manuscript. We will also mention that the information from literature in relation to this subject is very different function of the nature/size of ZnO and its surface treatments, the type of polymer matrix, processing methods, thickness of films and their surface, etc., thus the T values and reported results regarding the transmission of light can be very distinct. However, it is very clear that there is a great consensus regarding the absorption of UV light by ZnO particles, while it was more difficult to explain the decrease of T in visible region with increasing ZnO (NPs) loading in various polymer matrices, taking in consideration that the thin films of ZnO and ZnO (NPs) are transparent (M. Berber et al. / Scripta Materialia 53 (2005) 547–551; Insoo Kim et al., Polymers 2019, 11, 1427; doi:10.3390/polym11091427). According to Marra et al. ((2017) Effect of TiO2 and ZnO on PLA degradation in various media. Adv Mater Sci 2: DOI: 10.15761/AMS.1000122)), the lower values of T shown by the ZnO composite films by increasing the ZnO amount is due to the absorption and reflection of the particles and agglomerates of the ZnO particles. We thank the Reviewer for this useful comment and the additional explanation regarding the decreasing of T in visible region.

The following paragraph was added in the revised manuscript:

“The optical loses and important reduction of transmittance for the films having higher nanofiller amount are probably due to the scattering of visible light, absorption and reflection effects of ZnO NPs and inherent ZnO clusters.”

R1. ."3. It should be specified in the caption which parameters Table 7 presents."

Authors. In relation to this comment, we believe that the structure of papers for Molecules (i.e., firstly the section “Results and Discussion”, then the section “Materials and Methods”) is not all the time very friendly with the Readers and Authors! The Experimental section could contain very important information for the understanding and discussion of results, thus in some cases will be ideally to place this information immediately after the introductive part. In fact, the abbreviation of parameters from Table 7 is given previously in the footer of Table 4, and then, in the section “3.5.3. Differential Scanning Calorimetry (DSC).” Moreover, following the suggestion of Reviewer, the following mention was added in the footer of Table 7:

“*Abbreviations: glass transition temperature (Tg), cold crystallization temperature (Tcc), enthalpy of cold crystallization (ΔHcc), melting temperature (Tm), melting enthalpy (ΔHm), degree of crystallization (χc)”

  1. The following expression “UV-vis spectra” in the text and caption of Figure 9 must be replaced with “UV-vis transmittance spectra”.”Authors: We thank for this remark, the correction was done in the revised manuscript (text, caption of Fig. 9) by specifying “UV-vis transmittance spectra”.

Finally, we would like to thank the Reviewer for the time spent in reading and reviewing our manuscript, for the appreciation of our manuscript, amenable comments, and suggestions. The paper has been fully revised according to all comments and all changes have been highlighted by yellow ink directly within the revised contribution.

Reviewer 2 Report

Dear Editor

I read your paper and here are my comments:

  1. You should better explain the novelty of this research and the added value to the field.
  2. How did you determine the Young modulus? I think you could add the plots for mechanical test as supplementary material.
  3. What about the surface wettability of the films? Did you perform contact angle measurements?
  4. The paper is well written with useful data for the field of polymer processing and various applications.

Author Response

Detailed Responses to Reviewers (Reviewer 2)

Reviewer #2 (R2):

“I read your paper and here are my comments:

  1. You should better explain the novelty of this research and the added value to the field.
  2. How did you determine the Young modulus? I think you could add the plots for mechanical test as supplementary material.
  3. What about the surface wettability of the films? Did you perform contact angle measurements?
  4. The paper is well written with useful data for the field of polymer processing and various applications.”:

Authors: Firstly, we would like to thank the Reviewer 2 (R2) for the general appreciation of our manuscript and positive comments. Please find below our point-by-point answers.

R2:” 1. You should better explain the novelty of this research and the added value to the field.”

Authors: In relation to this comment, we would like to precise that following an original research and the interest for new biomaterials, it was first reported by us (Biomacromolecules 2011, 12, 5, 1762–1771) the possibility to produce PLA - ZnO nanocomposites via melt-blending technology (process easy transferable at larger scale) and using ZnO NPs previously surface‐treated with a specific silane (i.e., triethoxycaprylylsilane) to limit polyester (PLA) degradation. Then, following the contacts with potential users, additional developments have been considered (e.g., the case of PLA – ZnO MBs for films, fibers, injection molded items) to produce more performant products and to evidence also their multifunctional end-use properties (please consider the references from manuscript realized by contribution of authors for this topic). Moreover, to answer at new requests/questions of both, academic and industrial interest, this study highlights the beneficial effects of MB addition on the processing (better rheology, melt-strength, etc.) and properties of PLA - ZnO nanocomposites, i.e., in the case of nanocomposite films as final products. The utilization of MBs can offer more flexibility to the industrial users in relation to the choice of ZnO loadings by considering the final requirements of application (i.e., UV shielding, antibacterial protection, others). Still, it is noteworthy mentioning that “the MB method offers to processors the possibility to use varied techniques of processing (extrusion, spinning, injection molding), cumulated benefits such as cost savings and technical performances, cleaner facilities, no health, and safety/pollution risks connected to the manipulation of nanofillers, and so on.”

For easy understanding by the Reader, and following the suggestion of Reviewer, we have modified and improved the paragraph which introduces the novelty of research and the added value to the field:

“Spectacular improvements can be obtained by limiting the residence time at high temperature of PLA in presence of ZnO. Unfortunately, according to the best of our knowledge, the potential of the masterbatch (MB) technique was not enough considered to produce these specific nanocomposites. Furthermore, the utilization of MBs could represent a more flexible way to obtain products having different loadings of ZnO, improved processing and thermo-mechanical properties, better preservation of the molecular weights. The objective of this study has been to highlight the beneficial effects of MB addition on the performances of PLA - ZnO nanocomposite films as final products, but other applications can be also concerned. Moreover, this is one of the first works dealing with the production and properties of PLA – ZnO MBs, while the characteristic features of final nanocomposites extruded as films using highly filled MBs are directly compared to those of similar materials produced using the traditional technology. Last of all, these developments are viewed as promising novel techniques that can open new perspectives for the application at larger scale of PLA - ZnO nanocomposites characterized by multifunctional end-use properties (UV protection, antibacterial, gas-barrier, etc.).”

R2: 2. How did you determine the Young modulus? I think you could add the plots for mechanical test as supplementary material.

Authors: We thank the Reviewer for this question and suggestion, which gives us the possibility to improve the manuscript and to introduce the tensile stress - strain curves in the supplementary material. As mentioned in the section “Characterization”, the tensile testing measurements were performed on rectangular strips (64 x 10 x ~0.4 mm3, cut from the extruded films) according to ASTM D 638-02a and using a Lloyd LR 10K tensile bench at a speed rate of 1 mm/min, with 25.4 mm distance between grips. All mechanical tests were carried out by using specimens previously conditioned for at least 48 hours at 20 ±2 °C under a relative humidity of 50 ±3 % and the values were averaged out over minimum five measurements. The NEXYGEN™ MT Materials Test and Data Analysis Software was used to configure the tests, analyze and report the results. The nominal strain was determined as the change in grip separation relative to the original grip separation expressed as a percent. The values of Young’s modulus were obtained directly via software by considering a) the secant modulus method at low deformation (between 0.05% and 0.25% strain) and b) the values of initial tangential modulus indicated by program. Similar results were obtained using both methods, the good accuracy being ascribed to the high elasticity/rigidity of these specific nanocomposites, in fact one key characteristic of PLA.

Following the comment of the Reviewer, additional information was introduced in the revised manuscript, while in the supplementary material, as suggested, are shown typical tensile stress - strain curves:

“The NEXYGEN™ MT Materials Test and Data Analysis Software was used to configure the tests, analyze and report the results. The nominal strain was determined as the change in grip separation relative to the original grip separation expressed as a percent. The values of Young’s modulus were obtained directly via software and validated for accuracy considering two methods: 1. the secant modulus at low deformation (between 0.05% and 0.25% strain) and 2. the initial tangential modulus (values reported as Young’s modulus).”

As it was suggested by Reviewer, the new Figure S1 was added in the supplementary material (please consider the PDF variant with our answers).

“Figure S1. Typical tensile stress – strain curves of PLA - ZnO nanocomposite films (thickness ~400 µm) produced using a) the MB approach (method A) and b) PLA – ZnO nanocomposites as granules (method B, current technique)

Figure S1 (a, b). Typical tensile stress – strain curves of PLA - ZnO nanocomposite films (thickness ~400 µm) produced using a) the MB approach (method A) and b) PLA – ZnO nanocomposites as granules (method B, current technique)

R2. 3. What about the surface wettability of the films? Did you perform contact angle measurements?

Authors: Unfortunately, up to now some properties of PLA – ZnO nanocomposites (wettability, contact angle measurements, others) were not determined by us or our collaborators, the focus being mostly on the technology of production. Following this question, we also assume that could be of interest to have information about the CA values and antibacterial properties, thus a forthcoming study can be considered.

However, in relation to this question, some authors have been reported improved hydrophobicity and the increasing of water contact angle (WCA) of  PLA – ZnO films compared to the neat PLA films at increased ZnO loading (e.g., Insoo Kim et al., Polymers 2019, 11, 1427; doi:10.3390/polym11091427; Lizundia, E., et al. (2016), Poly(l‐lactide)/ZnO nanocomposites as efficient UV‐shielding coatings for packaging applications. J. Appl. Polym. Sci., 133, 42426, doi: 10.1002/app.42426).

Some similar results, i.e., the increase of WCA with ZnO loading can be expected in our case, by also considering the surface treatment of ZnO NPs with a specific silane having lipophilic caprylyl groups. We thank the Reviewer for the suggestion regarding investigation of WCA which can be concerned in the frame of a further/planned study regarding the realization of PLA nanocomposites characterized by self-cleaning properties, an occasion to perform these specific measurements.

R2: “4. The paper is well written with useful data for the field of polymer processing and various applications.”:

Authors: We would like to thank the Reviewer for the time spent in reading and reviewing our manuscript, for the appreciation of our manuscript, amenable comments, and suggestions. The paper has been fully revised according to all comments and all changes have been highlighted by yellow ink directly within the revised contribution.
